# DrugCLIP: Contrastive Protein-Molecule Representation Learning for Virtual Screening

**Bowen Gao[1]\*, Bo Qiang[2]\*, Haichuan Tan[1], Minsi Ren[3], Yinjun Jia[4], Minsi Lu[5],**
**Jingjing Liu[1], Wei-Ying Ma[1], Yanyan Lan[1,6]†**

[1]Institute for AI Industry Research (AIR), Tsinghua University
[2]Department of Pharmaceutical Science, Peking University
[3]Institute of Automation, Chinese Academy of Sciences
[4]School of Life Sciences, Tsinghua University
[5]Department of Pharmaceutical Science, Tsinghua University
[6]Beijing Academy of Artificial Intelligence

## Abstract

Virtual screening, which identifies potential drugs from vast compound databases to bind with a particular protein pocket, is a critical step in AI-assisted drug discovery. Traditional docking methods are highly time-consuming, and can only work with a restricted search library in real-life applications. Recent supervised learning approaches using scoring functions for binding-affinity prediction, although promising, have not yet surpassed docking methods due to their strong dependency on limited data with reliable binding-affinity labels. In this paper, we propose a novel contrastive learning framework, DrugCLIP, by reformulating virtual screening as a dense retrieval task and employing contrastive learning to align representations of binding protein pockets and molecules from a large quantity of pairwise data without explicit binding-affinity scores. We also introduce a biological-knowledge inspired data augmentation strategy to learn better protein-molecule representations. Extensive experiments show that DrugCLIP significantly outperforms traditional docking and supervised learning methods on diverse virtual screening benchmarks with highly reduced computation time, especially in zero-shot setting. The code for DrugCLIP is available at `https://github.com/bowen-gao/DrugCLIP`.

## 1 Introduction

Virtual screening is a crucial computer-aided drug discovery (CADD) technique that uses computational methods [26] to search for candidate drug structures from compound libraries, aiming to identify molecules most likely to bind to a target (*e.g.*, a protein receptor or enzyme). Common practice follows the wisdom of "bigger is better" [25]: the larger the search library, the better chance to find a matching drug candidate. For example, statistics show that increasing the data size from $10^5$ to $10^8$ leads to a significant jump in the number of true ligands among top-1000 results [19].

Molecular docking [11, 43, 41] is currently the dominant virtual screening method, which models protein-molecule interaction by a quantitative score correlated with the free energy of binding. The computation of such scores heavily depends on molecule orientations and conformations sampling, which is time-consuming and impractical when dealing with large libraries in the magnitude of billions [14]. As estimated by [32], given a standard computating rate of 10 seconds per compound on

---

\*Equal contirbution
†Correspondence to `lanyanyan@air.tsinghua.edu.cn`

a single CPU core, it takes 3000 years to complete screening 10 billion compounds using commercial docking methods (costing over 800k dollars). Consequently, docking methods are gravely limited by their high computational cost and slow inference speed.

As molecule libraries continue to grow, new high-throughput virtual screening methods are in pressing demand. Supervised learning algorithms [20] such as regression [22] and classification [54] have been investigated for binding-affinity prediction. However, these supervised methods usually require carefully labeled data samples for training, thus struggling with poor generalization [44]. In addition, the scarcity of reliable negative samples imposes restrictions on model performance. As a result, they still underperform current docking methods.

Rethinking the problem of virtual screening, we find that the key issue is to identify which molecules are likely to bind with a protein pocket, instead of determining the accurate binding-affinity score (the goal of prediction in regression/classification models) or binding pose (the goal of docking). Following this thought, we recast virtual screening as an information retrieval task, *i.e.*, given a protein pocket as the query, we aim to retrieve from a large-scale molecule library the most relevant molecules with the highest probability of binding to the target pocket. In this new perspective, virtual screening is boiled down to a similarity matching problem between proteins and molecules.

To this end, we introduce DrugCLIP, a dense retrieval approach (inspired by CLIP [31]) that computes a contrastive loss between two separate pre-trained encoders to maximize the similarity between a protein-molecule pair, if they have a binding affinity, and minimize it otherwise. Compared with supervised learning methods, our contrastive learning approach enjoys several advantages. Firstly, the objective of finding the matching relations between proteins and molecules is naturally in accordance with the formulation of the virtual screening task. Secondly, the designed contrastive loss relieve the dependency on explicit labeling of binding affinity, and facilitates the usage of large-scale unlabeled data beyond densely annotated small datasets such as PDBBind [45]. Thus, we extend the ability of our model with a large pool of protein-molecule pairs by utilizing BioLip [47] and ChEMBL [8] datasets for training. We further introduce a biological-knowledge inspired augmentation method, *HomoAug*, which creates protein-molecule pairs based on protein homology evolutions. Lastly, the dense retrieval setting allows for offline pre-computation of protein and molecule encodings, bringing high efficiency to online inference and promising high-throughput virtual screening on billions of molecules.

Experiments on two challenging virtual screening benchmarks, DUD-E [28] and LIT-PCBA [40], demonstrate that zero-shot performance of our model surpasses most deep learning baselines that carefully finetune on labeled data. We also conduct a human evaluation to compare DrugCLIP with Glide, a commercial docking system widely used by pharmacology experts. In 80% cases, judges prefer the selection of top-10 molecules from our method over Glide. Furthermore, since DrugCLIP indiscriminately models the similarity between protein and molecule, it can be extended to other important tasks in drug discovery such as 'target fishing', where protein candidates are ranked for a given molecule. DrugCLIP also outperforms docking methods on these benchmarks.

Our main contributions are summarized as follows:

- To our best knowledge, this is the first effort to position large-scale virtual screening as a dense retrieval problem, which enables ultra-fast screening over billion-scale chemical libraries for candidate search by storing pre-computed molecule embeddings offline.
- We propose a novel contrastive learning framework that learns a generic joint representation of proteins and molecules, which can be applied to molecule-pocket pairing tasks. Novel data augmentation strategy and training techniques are also introduced.
- DrugCLIP, with its impressive zero-shot performance on virtual screening benchmarks, well addresses poor generalization and low efficiency issues faced by docking and learning-based screening methods.

## 2   Related Work

There are mainly two schools of virtual screening methods, molecular docking and supervised learning. Molecular docking is a computational technique that predicts the binding energy, optimal orientation, and conformation of a small molecule ligand within a protein binding site [19]. It uses sampling algorithms such as genetic algorithms [41] and Monte Carlo [11, 43] to generate a set of

candidate ligand poses, by exploring the conformational space of the ligand and the protein receptor. These candidate poses are then evaluated by molecule-protein scoring functions such as empirical force fields [21] to assess their binding affinity. This iterative process continues until convergence, which is computationally demanding.

To accelerate the prediction process, supervised learning methods have emerged as an alternative to the iterative refining process in docking. A recent work [13] proposes pocket pretraining to find ligands for similar pockets by only utilizing information from one side of the pocket-ligand pairs. By training on given binding-affinity labels, regression models such as DeepDTA [30], OnionNet [52], GraphDTA [29] and SG-CNN [16] learn the mapping between protein-molecule representations by first predicting the binding affinity for every protein-molecule pair, then ranking them to determine top candidates. However, these models suffer from high false-positive rates, due to the lack of negative binding-affinity data. Another way is to use predefined rules (*e.g.*, DUD-E [28]) to obtain negative samples and train a classifier to discriminate positive and negative protein-molecule pairs (DrugVQA [54], AttentionSiteDTI [24]). Previous work has observed poor generalization in these methods. For example, Wang and Dokholyan [44] suggests that models trained on DUD-E [28] cannot be transferred to other classification benchmarks.

Concurrently, several studies have employed contrastive learning objectives in the context of virtual screening. Singh et al. [36] utilized a protein language model and rule-based molecule fingerprints for representation, while CoSP [7] opted for chemical similarity in their negative sampling approach. In the meantime, we propose a more holistic and flexible solution by leveraging an equivariant 3D model, efficient dense retrieval, and novel data augmentation techniques.

## 3 DrugCLIP Framework

### 3.1 Overview

To formulate the problem, we denote the protein pocket of interest as $p$, and a set of $n$ small molecules is represented by $\mathcal{M} = \{m_1, m_2, \ldots, m_n\}$. The objective of virtual screening is to identify the top $k$ candidates with the highest probability of binding to the target pocket. This selection process is typically guided by a scoring function $s(\cdot, \cdot)$, which assesses the pairwise data between the pocket $p$ and each candidate molecule $m_i$. The scoring function can be derived from techniques such as docking simulations or supervised learning models, which perform ranking and selection of the most promising candidates based on their likelihood of binding to the target pocket.

To view virtual screening as a dense retrieval task, we treat the pocket as the query to retrieve relevant molecules from the given library. The overall framework is illustrated in Figure 1. First, two separate encoders are trained to learn the representations of protein pockets (abbreviated as proteins hereafter) and molecules. Then, the similarity between each protein-molecule pair is computed and a contrastive learning objective is utilized to discriminate between positive and negative pairs. All the parameters in the encoders and similarity functions are trained jointly.

### 3.2 Protein and Molecule Encoders

Diverse representation learning methods can be used as protein and molecule encoders. In this paper, we follow the encoder architecture of UniMol [55], a powerful 3D encoder pre-trained with large-scale unsupervised data. Here we briefly introduce the encoding process.

Firstly, both molecules and protein pockets are tokenized to atoms. A molecule with $L$ tokens is denoted as a feature vector $x^m = \{c_m, t_m\}$, where $c_m \in \mathbb{R}^{L \times 3}$ represents the atom coordinates and $t_m \in \mathbb{R}^L$ represents the atom types. The same setting is applied to obtain pocket features, denoted as $x^p = \{c_p, t_p\}$.

As described in UniMol [55], the encoder is a SE(3) 3D transformer that accepts tokenized atom features as input. To preserve SE(3) invariance for embedding the molecular structure, the 3D coordinate features are utilized as geometric distances. Specifically, the pairwise representation $q_{ij}^0$ is initialized based on the distance between each pair of atoms. For each transformer layer $l$, the self-attention mechanism for learning atom representation is defined in Equation 1. The pairwise representation serves as a bias term in the attention mechanism, encoding 3D features into atom representations. The update rules between adjacent transformer layers are also defined in Equation 1.

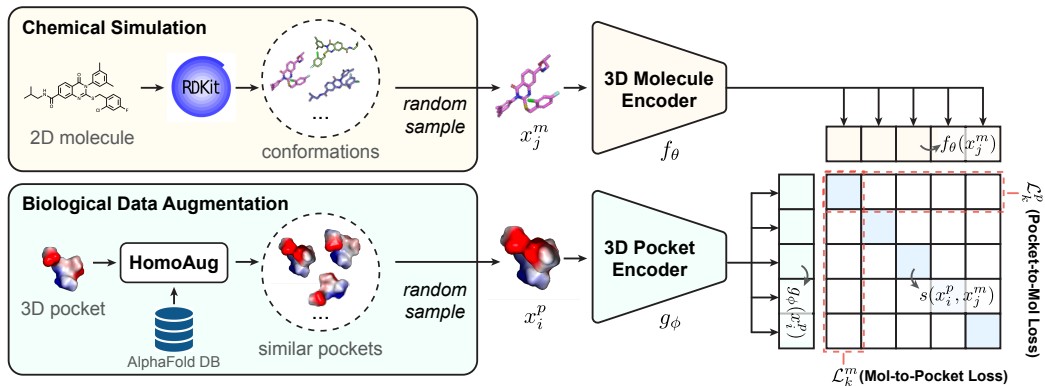

Figure 1: An illustration of the training procedure. Molecule conformations are generated by RDkit simulation, and pocket data are augmented with HomoAug. At each training iteration, sampled 3D molecule and 3D pocket representations are learned with a contrastive objective.

$$\text{Attention}(Q_i^l, K_j^l, V_j^l) = \text{softmax}(\frac{Q_i^l(K_j^l)^T}{\sqrt{d}} + q_{ij}^l)V_j^l, \quad \text{where } q_{ij}^{l+1} = q_{ij}^l + \frac{Q_i^l(K_j^l)^T}{\sqrt{d}}. \quad (1)$$

Inspired by BERT[5], we randomly mask atom types and pretrain our model by predicting the masked atom types. Besides, we introduce random noise to corrupt atom coordinates and pretrain the model to reconstruct the original coordinates. Specifically, uniform noises of [-1, 1] are added to 15% of atom coordinates, where the pair-distance prediction heads estimate uncorrupted distances and the SE(3)-equivariant head directly predicts correct coordinates. Detailed implementations are described in the Appendix B. A special atom [CLS] with coordinates at the center of all atoms is added to output the representation of the corresponding protein and molecule. Specifically, we denote the protein encoder as $g_\phi$ and molecule encoder as $f_\theta$. Then, the representation of protein $x^p$ and molecue $x^m$ are defined as $g_\phi(x^p)$ and $f_\theta(x^m)$, correspondingly.

### 3.3 Contrastive Learning Objective

To conduct the contrastive learning process, we first need to obtain the similarity measurements between each protein-molecule pair. Following previous work, both dot product and cosine similarity can be adopted as the similarity functions. For example, when using dot product, the similarity score of $(x_i^p, x_j^m), \forall i, j \in [1, N]$ can be written as:

$$s(x_i^p, x_j^m) = g_\phi(x_i^p)^T \cdot f_\theta(x_j^m), \quad (2)$$

while cosine similarity can be obtained by normalization.

In the field of virtual screening, positive pairs of binding protein and molecule are usually provided, with limited true negative pairs. Therefore, we need to construct negative pairs for the contrastive objective. Here we use an in-batch sampling strategy similar to CLIP [31]. Specifically, given a batch of paired data $\{(x_k^p, x_k^m)\}_{k=1}^N$ with batch size $N$, we extract a list of proteins $\{x_k^p\}_{k=1}^N$ and a list of corresponding molecules $\{x_k^m\}_{k=1}^N$. Combining them together results in $N^2$ pairs $(x_i^p, x_j^m)$ where $i, j \in [1, N]$. When $i = j$ it is a positive pair, and when $i \neq j$ it is a negative pair.

Please note that the in-batch negative construction is intrinsically based on a simple assumption that, if a certain pair of protein and molecule has been tested as having a binding relation, it is likely that they have a negative binding relation with other molecule/protein. This assumption is reasonable as the true distribution of positive and negative molecules exhibits a sharp contrast, with a proportion significantly smaller than 0.1% [25].

Formally, we introduce two losses: Pocket-to-Mol loss and Mol-to-Pocket loss. The former describes the likelihood of ranking its binding molecules before other molecules for a given protein $x_k^p$:

$$\mathcal{L}_k^p(x_k^p, \{x_i^m\}_{i=1}^N) = -\frac{1}{N} \log \frac{\exp(s(x_k^p, x_k^m)/\tau)}{\sum_i \exp(s(x_k^p, x_i^m)/\tau)}, \quad (3)$$

while the latter is the likelihood of ranking its binding targets for a given molecule $x_k^m$, and is defined as:

$$\mathcal{L}_k^m(x_k^m, \{x_k^p\}_{i=1}^N) = -\frac{1}{N} \log \frac{\exp(s(x_k^p, x_k^m)/\tau)}{\sum_i \exp(s(x_i^p, x_k^m)/\tau)}. \tag{4}$$

In the above two equations, $\tau$ represents the temperature parameter that controls the softmax distribution, which has been widely utilized in previous representation learning methods[3, 10, 46, 31].

Combining the two losses, we obtain the final loss for a mini-batch:

$$\mathcal{L} = \frac{1}{2} \sum_{k=1}^N (\mathcal{L}_k^p + \mathcal{L}_k^m). \tag{5}$$

### 3.4 Training and Inference

Virtual screening with DrugCLIP contains two phases. In the offline phase, embeddings of each molecule are obtained by DrugCLIP encoders $f_\theta$. These embedding vectors are then stored in memory for later-stage online retrieval. Specifically, for a given query protein pocket, it is first encoded into an embedding vector using the trained protein encoder $g_\phi$. We then measure the similarity between the encoded pocket vector and all the embedding vectors of candidate molecules (dot product or cosine similarity). Finally, we proceed to select the top-$k$ molecules from the candidate pool based on their similarity scores. Notably, our method offers a distinct advantage compared to other supervised learning screening frameworks. While other methods involve complex neural network computations for the scoring functions during the online screening phase, our approach capitalizes on the pre-computed and cached candidate embedding vectors. Consequently, the only computation required is the high-speed dot product calculation. This novel design allows for rapid screening of a large number of candidates, without incurring additional computational overhead. For a detailed time analysis, please refer to Section 4.3.

**Constructing Training Data**

We use three datasets for training: PDBBind [45], BioLip [47], and ChEMBL [8]. PDBbind is a standard database used in docking and binding-affinity prediction. It consists of experimentally measured protein-ligand complex structures along with their binding-affinity labels, from which true positive protein-molecule pairs with accurate structures can be extracted. We use PDBBind 2019, which includes over 17,000 protein-molecule complexes with binding-affinity data covering a wide range of chemical space and protein families. We use the general set for training and the refined set for validation.

BioLip is a dataset updated weekly by a standard data mining workflow that extracts complex structure data from PDB. We filter out all complexes that contain peptides, DNA, RNA, and single ions, and obtain 122861 protein-molecule pairs, much larger than PDBBind.

DrugCLIP model can also use known receptor-ligand pairs without their binding structures. From the ChEMBL [8] dataset, we filter out proteins with only one known binding pocket. Then we pair the pocket with all positive binders in the ChEMBL database. We hypothesize (supported by domain experts [9, 18]) that assayed ligands dominantly bind to the known pocket in solved structures, and our model can tolerate introduced noise of this filtration protocol.

Since our deep-learning-based method is insensitive to occasional inaccuracies in coordinates and can be trained with raw element types. Therefore, only minimal cleaning-ups are performed for the protein structures in the above mentioned datasets to remove irrelevant molecules like water.

**Biological Data Augmentation**

Directly applying common data augmentation techniques to augment biological data is infeasible, as introducing noise or perturbations to pocket or molecule data can result in unstable or chemically incorrect structures, rendering the augmented data unreliable, especially for virtual screening [34]. To address this challenge, we propose a new augmentation method called HomoAug that takes into account the biological significance of the data. It utilizes the concept of homologous proteins in biology, to combine ligands from PDBbind [45] with homologous proteins corresponding to their pockets, thereby generating new training data (Figure 2).

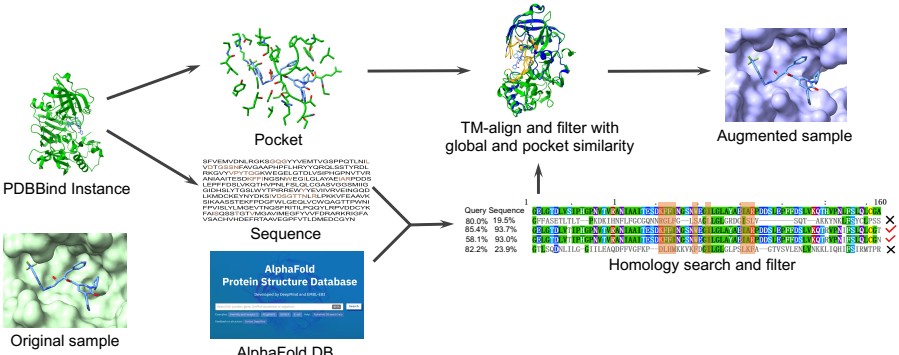

Figure 2: Illustration of HomoAug's pipeline. The pocket protein instances from PDBBind are searched for homologous counterparts in the AlphaFold Protein Structure Database. Then the TMalign method is employed to achieve structural alignment between the homologous protein and the original protein. Following a filtering process, the homologous protein is combined with the ligand to create an augmented pocket-ligand pair.

**Training-test Inconsistency**

There exists a training-test inconsistency problem in virtual screening. This is because holo structures in training data are depicted after binding, which differs from their apo structures. Previous methods usually use docking software to obtain estimated binding structure. For better efficiency, we propose to use the chemical simulation package [23], RDkit to generate noisy data as input. For a given protein $x^p = \{c_p, h_p\}$ and molecule $x^m = \{c_m, h_m\}$, we denote $c$ as coordinates and $h$ as atom types. Inaccurate molecule coordinates $c_m + \delta$ simulated by RDkit are used as noisy input $\tilde{x}_m = \{c_m + \delta, h_m\}$. It should be noted that our model adopts a dual-tower architecture, while previous 3D supervised learning methods are mainly based on a single tower [52, 16]. The following Proposition proves that the scoring function of DrugCLIP is more robust than those obtained by supervised learning methods. In a more general setting, we suggest that dual-tower models based on 3D encoders enjoy more generalization by decoupling the relative 3D distances from the prediction. By learning molecule-protein interactions from noisy unbind molecule conformations, our model addresses the consistency issue between training and testing.

**Proposition 1.** *Denote the scoring function of a supervised learning method as $k_\gamma$, we have,*

$$\lim_{\delta \to 0} \{s(\tilde{x}_m, x_p) - s(x_m, x_p)\} = 0, \tag{6}$$

$$\lim_{\delta \to 0} \{k_\gamma(h_p, h_m, c_p, \tilde{c}_m) - k_\gamma(h_p, h_m, c_p, c_m)\} \neq 0. \tag{7}$$

*The proof is provided in the Appendix A*

## 4 Experiments

We first introduce evaluation metrics. Although AUROC (area under the receiver operating characteristic curve) is commonly used for classification, it has been criticized for being unsuitable for virtual screening. [51] This is because the target of virtual screening is to select a small fraction of molecules from a large pool,resulting in a significantly low false positive rate (FPR) in this scenario. However, AUROC is calculated by averaging the FPR from 0 to 1. To overcome this, we also use BEDROC (Boltzmann-enhanced discrimination of ROC), Enrichment Factor(EF) and ROC enrichment metric (RE) for evaluation. BEDROC incorporates exponential weights that assign greater importance to early rankings. EF and RE are two widely used metrics for virtual screening (detailed definitions in Appendix B).

### 4.1 Evaluation on DUD-E Benchmark

DUD-E [28] is one of the most popular virtual screening benchmarks. It contains 102 proteins with 22,886 bio-active molecules, for which 50 topological dissimilar decoys that possess matched physicochemical properties are retrieved from the ZINC database.

Table 1: Results on DUD-E in zero-shot setting.

| | AUROC (%) | BEDROC (%) | EF | | |
| --- | --- | --- | --- | --- | --- |
| | | | 0.5% | 1% | 5% |
| Glide-SP [11] | 76.70 | 40.70 | 19.39 | 16.18 | 7.23 |
| Vina [41] | 71.60 | - | 9.13 | 7.32 | 4.44 |
| NN-score [6] | 68.30 | 12.20 | 4.16 | 4.02 | 3.12 |
| RFscore [1] | 65.21 | 12.41 | 4.90 | 4.52 | 2.98 |
| Pafnucy [38] | 63.11 | 16.50 | 4.24 | 3.86 | 3.76 |
| OnionNet [52] | 59.71 | 8.62 | 2.84 | 2.84 | 2.20 |
| Planet [49] | 71.60 | - | 10.23 | 8.83 | 5.40 |
| DrugCLIP$_{ZS}$ | **80.93** | **50.52** | **38.07** | **31.89** | **10.66** |

Table 2: Results on DUD-E in finetuning setting.

| | AUROC (%) | RE | | | |
| --- | --- | --- | --- | --- | --- |
| | | 0.5% | 1% | 2% | 5% |
| COSP[7] | 90.10 | 51.05 | 35.98 | 23.68 | 12.21 |
| Graph CNN[39] | 88.60 | 44.41 | 29.75 | 19.41 | 10.74 |
| DrugVQA[53] | **97.20** | 88.17 | 58.71 | 35.06 | **17.39** |
| AttentionSiteDTI[48] | 97.10 | 101.74 | 59.92 | 35.07 | 16.74 |
| DrugCLIP$_{ZS}$ | 80.93 | 73.97 | 41.79 | 23.68 | 11.16 |
| DrugCLIP$_{FT}$ | 96.59 | **118.10** | **67.17** | **37.17** | 16.59 |

In the zero-shot setting, we compare with docking and other learning methods. Since all learning methods only use PDBBind for training, we also train our model, named DrugCLIP$_{ZS}$, on PDBBind for fair comparison. We exclude all the targets present in DUD-E from our training set to ensure zero-shot learning. For the fine-tuning setting, further tuning on DUD-E is required. We follow the same split and test approach in Jones et al. [16]. The finetuned model is named DrugCLIP$_{FT}$.

Table 1 and 2 summarize the results in zero-shot and fine-tuning setting, respectively. From Table 1, we can see that our model outperforms both docking and learning methods in zero-shot setting by a large margin. Besides, our model is the only one outperforming traditional molecule docking methods. As for the comparison in finetuning setting, DrugCLIP$_{FT}$ model, although achieving a lower AUROC compared to other finetuned models, outperforms them in terms of RE at 0.5%, 1%, and 2% levels. This indicates that our model is particularly well-suited for virtual screening tasks that prioritize the identification of hit molecules at a small fraction of the entire dataset. Surprisingly, we found that even DrugCLIP$_{ZS}$ outperforms some of the supervised-learning methods. These results show that DrugCLIP harnesses great strengths, especially in zero-shot setting, which approximates virtual screening in real-world applications.

## 4.2 Evaluation on LIT-PCBA Benchmark

LIT-PCBA is a much more challenging virtual screening benchmark, proposed to address the biased data problem faced by other benchmarks, *e.g.*, DUD-E. Based on dose-response PubChem bioassays, the LIT-PCBA dataset consists of 15 targets and 7844 experimentally confirmed active and 407,381 inactive compounds.

For fair comparison, we also used PDBBind as training data. Since all baselines are in a zero-shot setting, we exclude all the targets present in LIT-PCBA from our training set.

As shown in Table 3, DrugCLIP consistently outperforms commercial docking methods (Surflex and Glide-SP). Despite not achieving the highest AUROC, DrugCLIP excels in the more critical BEDROC and EF scores for virtual screening, surpassing all other baselines by a large margin. Additionally, all methods demonstrate lower performances on LIT-PCBA compared to DUD-E, indicating the greater challenge posed by LIT-PCBA for virtual screening.

Table 3: Results on LIT-PCBA.

|  | AUROC (%) | BEDROC (%) | EF | | |
|---|---|---|---|---|---|
|  |  |  | 0.5% | 1% | 5% |
| Surflex [37] | 51.47 | - | - | 2.50 | - |
| Glide-SP [11] | 53.15 | 4.00 | 3.17 | 3.41 | 2.01 |
| Planet [49] | 57.31 | - | 4.64 | 3.87 | 2.43 |
| Gnina [27] | **60.93** | 5.40 | - | 4.63 | - |
| DeepDTA [30] | 56.27 | 2.53 | - | 1.47 | - |
| BigBind [2] | 60.80 | - | - | 3.82 | - |
| DrugCLIP | 57.17 | **6.23** | **8.56** | **5.51** | **2.27** |

## 4.3 Efficiency Analysis for Large-scale Virtual Screening

DrugCLIP offers a significant advantage in terms of ultra-fast speed. To evaluate it on real-world databases, we analyze two scenarios: performing one-time virtual screening with a specific target on various libraries, and conducting multi-time screening with multiple targets on a fixed library. Here we use Planet [49] as a representative of ML scoring function (MLSF) based supervised learning methods.

For the first scenario, as illustrated in Figure.3a, when all candidate molecules are not pre-encoded, our method requires a comparable amount of time to other learning methods. However, if all candidate molecules are pre-encoded into embeddings and stored in memory, DrugCLIP can perform virtual screening in less than 10,000 seconds (approximately 30 hours) for Enamine, which comprises 6 billion molecules. This significant reduction in time demonstrates the efficiency and scalability of our method when leveraging pre-encoded molecules.

Results for the second scenario are presented in Figure.3b. Here, the search library is fixed and all molecules are pre-encoded and stored. When there are only 10 targets, the time difference between DrugCLIP and Planet [49] is approximately 10 days, which is manageable. However, as the number of targets increases to 600, the time difference expands to 2 years. These findings highlight the scalability challenge faced by existing learning-based screening methods when dealing with a large number of targets, and the huge efficiency advantage of DrugCLIP.

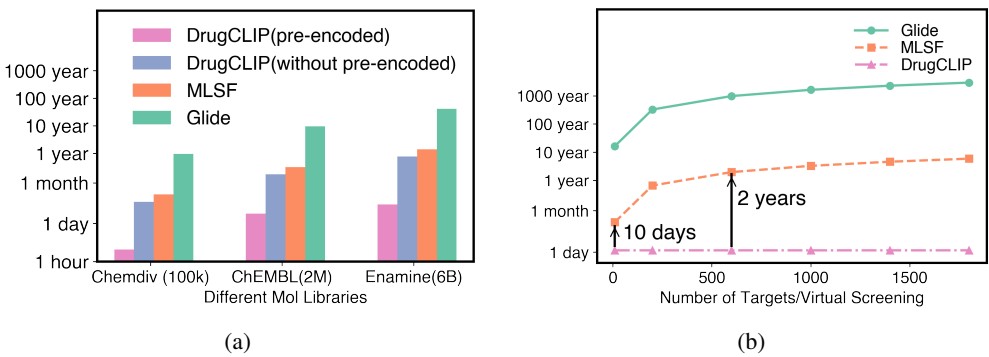

(a)                                                                 (b)

Figure 3: Time analysis of virtual screening with DrugCLIP, Glide, and a representative learning method. (a) virtual screening time on a single target with different molecule libraries. (b) virtual screening time on multiple targets with a fixed molecule library.

## 4.4 Ablation Studies

We conduct an ablation study to evaluate the two training techniques: employing HomoAug for data augmentation, and utilizing RDkit conformations to replace the original molecule conformation. Results are summarized in Table 5. We can see that performance improves by adding each of these techniques.

To clarify whether the surprisingly good results of DrugCLIP are attributed to the contrastive learning modules, or instead to the pre-trained encoder of UniMol, we further introduce DrugBA, a regression

Table 4: Ablation studies on different objectives.

| | DUD-E | | | LIT-PCBA | | |
|---|---|---|---|---|---|---|
| | AUROC % | BEDROC % | EF@1% | AUROC % | BEDROC % | EF@1% |
| DrugBA | 69.53 | 11.16 | 5.88 | 54.23 | 2.28 | 2.02 |
| DrugCLIP | 80.93 | 50.52 | 31.89 | 57.17 | 6.23 | 5.51 |

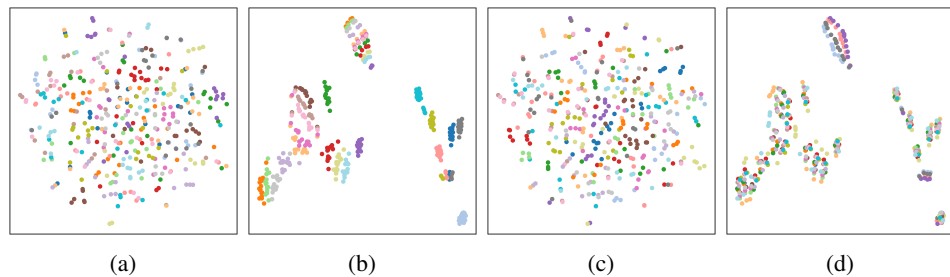

| (a) | (b) | (c) | (d) |

Figure 4: TSNE visualization of embeddings generated by combining pocket-molecule pairs. (a) and (c) represent embeddings produced by DrugCLIP, with (a) colored according to molecules and (c) colored according to pockets. (b) and (d) display embeddings generated by *DrugBA*, with (b) colored based on molecules and (d) colored based on pockets.

model utilizing the same encoder architecture as DrugCLIP. We train both models on PDBBind, and report their results in zero-shot setting, on both DUD-E and LIT-PCBA. Table 4 shows that DrugBA consistently underperforms DrugCLIP. Especially, if we compare the results with Table 1 and 3, DrugBA even underperforms existing docking methods. This further demonstrates the necessity and power of contrastive learning.

Table 5: Ablation results.

| Ablation Settings | | AUROC % | BEDROC % | EF | | |
|---|---|---|---|---|---|---|
| Rdkit Confs | HomoAug | | | 0.5% | 1% | 5% |
| ✗ | ✗ | 77.10 | 38.13 | 29.84 | 23.86 | 8.69 |
| ✓ | ✗ | 81.05 | 45.81 | 35.91 | 29.12 | 10.04 |
| ✓ | ✓ | 80.93 | 50.52 | 38.07 | 31.89 | 10.66 |

To visually demonstrate the disparity between binding-affinity prediction and contrastive learning, Figure 4 presents a visualization of the embeddings of DrugCLIP and DrugBA. For this visualization, 20 molecules and 20 pockets are randomly chosen from the CASF-2016 dataset. By combining these pairs using multiplication, we obtain 400 combined embeddings for further analysis. When the embeddings are labeled and colored by molecule index or pocket index, we can see that the embeddings produced by DrugBA exhibit a clustering pattern, suggesting that the model tends to assign similar scores to different pockets for a given molecule. In contrast, the embeddings produced by DrugCLIP exhibit no clustering, highlighting its ability to learn meaningful embeddings and mitigate the serious spurious bias in traditional binding-affinity prediction objectives. Thus, DrugCLIP is also able to outperform docking methods in target fishing task, which aims to find relative pockets given a specific molecule. Detailed results are in Appendix C.

### 4.5 Human Evaluation

We conduct a human evaluation experiment to compare our method with the most widely used screening software, Glide, to test DrugCLIP as a useful tool for pharmacological experts. We first conduct an experiment on exploring the limits of DrugCLIP, by comparing versions of DrugCLIP trained on different datasets. Model trained on PDBBind is named DrugCLIP-S, and similarly DrugCLIP-M and DrugCLIP-L for models trained on BioLip [47] and ChEMBL [8], respectively. All three models are augmented with PDBBind.

Figure 5 shows that DrugCLIP-M performs the best on DUD-E, while DrugCLIP-L is the best on LIT-PCBA. Considering that the setting of LIT-PCBA is much more challenging and real, we choose DrugCLIP-L for human evaluation (denoted as DrugCLIP in following discussion for consistency).

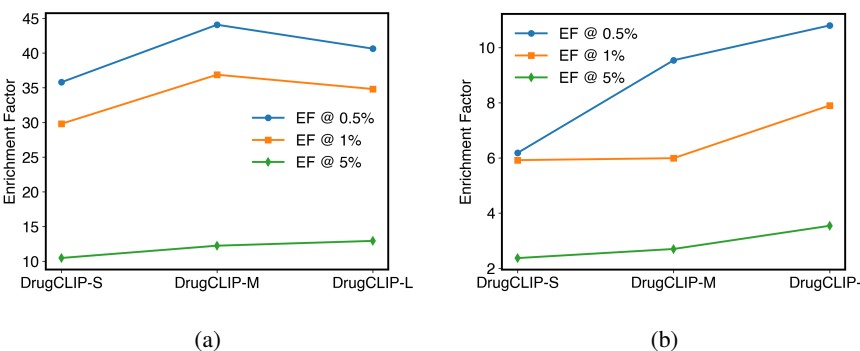

|          |          |
|:--------:|:--------:|
| (a)      | (b)      |

Figure 5: Comparison of training with different dataset versions on (a) DUD-E and (b) LIT-PCBA.

We select five targeting protein pockets and perform virtual screening with both Glide and DrugCLIP on the ChemDiv compound database for each target of interest. Then, the top-50 molecules from each method were selected and shuffled to create a list of 100 molecules. Domain experts who have drug design experience for specific targets are required to independently choose top-10 molecules from each 100 molecules. The results show that domain experts are inclined to choose the candidate structures given by DrugCLIP rather than Glide in 4 out of 5 cases, indicating great potential of DrugCLIP as an useful tool for human experts. Detailed results are reported in Appendix C.

## 5 Conclusion and Future Work

In this paper, we introduce "DrugCLIP", a novel approach for efficient and accurate virtual screening. Our method leverages contrastive learning to align the representations of binding pockets and molecules. By achieving state-of-the-art results and surpassing docking methods across diverse virtual screening benchmarks and tasks, DrugCLIP not only improves screening accuracy but also significantly reduces the time required for large-scale virtual screening. This opens up the possibility of expanding the search library to billions of compounds. There are several avenues for future research, such as designing further data augmentation techniques and investigating the use of more detailed atom-level interactions, which is explored in Appendix C.

## Acknowledgments and Disclosure of Funding

This work is supported by National Key R&D Program of China No.2021YFF1201600, Vanke Special Fund for Public Health and Health Discipline Development, Tsinghua University (NO.20221080053), and Beijing Academy of Artificial Intelligence (BAAI).

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

# A    Proof for Proposition

*Proof.* of Proposition 1 In order to ensure SE(3) equivariance in the encoder architecture, we define an additional function to derive the relative distance matrix of two coordinate systems, $c_x$ and $c_y$. Let $c_x = (x_1, y_1, z_1), (x_2, y_2, z_2), \ldots, (x_n, y_n, z_n)$ be the coordinates of system $c_x$, and let $c_y = (x'_1, y'_1, z'_1), (x'_2, y'_2, z'_2), \ldots, (x'_n, y'_n, z'_n)$ be the coordinates of system $c_y$. Define a function $D(c_x, c_y) = (d_{11}, d_{12}, d_{13}), (d_{21}, d_{22}, d_{23}), \ldots, (d_{n1}, d_{n2}, d_{n3})$ where $d_{ij} = \sqrt{(x_i - x'_j)^2 + (y_i - y'_j)^2 + (z_i - z'_j)^2}$ for $i, j = 1, 2, \ldots, n$. Since most popular SE(3) frameworks [33, 4, 35] utilize the relative distance to represent coordinates, we replace all coordinates with this matrix representation.

The deviation between using accurate ligand coordinates and inaccurate ligand coordinates can be written as $s(\tilde{x}_m, x_p) - s(x_m, x_p)$. If we applied the Taylor expansion of the first order, the deviation becomes proportionate to the distance perturbation.

$$
\begin{aligned}
& s(\tilde{x}_m, x_p) - s(x_m, x_p) \\
&= f_\theta \left( D\left(c_m + \delta, c_m + \delta\right), h_m \right)^\top g\left( D\left(C_p, C_p\right), h_p \right) - f_\theta \left( D\left(c_m, c_m\right), h_m \right)^\top g\left( D\left(C_p, C_p\right), h_p \right) \\
&\approx \frac{\partial f_\theta}{\partial D(c_m, c_m)} \left( D(c_m, c_m), h_m \right) \cdot \left( D\left(c_m + \delta, c_m + \delta\right) - D\left(c_m, c_m\right) \right)
\end{aligned}
\tag{8}
$$

If the RDkit simulated conformation of the ligand is close enough to the protein-induced conformation, we can find the optimal Rotation $R$ and translation $t$ to fit the two conformations in 3D space that satisfies $(c_n + \delta)R^\top + t = c_n$, which further means $D\left(c_m + \delta, c_m + \delta\right) - D\left(c_m, c_m\right) = 0$. Therefore, the deviation will be relatively small.

$$
\lim_{\delta \to 0} \{ s(\tilde{x}_m, x_p) - s(x_m, x_p) \} = 0
\tag{9}
$$

However, when we applied the first-order Taylor expansion to the deviation of the Single-Tower model, we find out that the deviation is not proportional.

$$
\begin{aligned}
& k_\gamma \left( h_p, h_m, D\left(c_p, c_p\right), D\left(c_m + \delta, c_m + \delta\right), D\left(c_m + \delta, c_p\right) \right) \\
& - k_\gamma \left( h_p, h_m, D\left(c_p, c_p\right), D\left(c_m, c_m\right), D\left(c_m, c_p\right) \right) \\
&\approx \frac{\partial k_\gamma}{\partial D\left(c_m + \delta, c_m + \delta\right)} (\cdot) \, \dot{} \left( D\left(c_m + \delta, c_m + \delta\right) - D\left(c_m, c_m\right) \right) \\
& + \frac{\partial k_\gamma}{\partial D\left(c_m, c_p\right)} (\cdot) \, \dot{} \left( D\left(c_m + \delta, c_p\right) - D\left(c_m, c_p\right) \right)
\end{aligned}
\tag{10}
$$

Though $D\left(c_m + \delta, c_m + \delta\right) - D\left(c_m, c_m\right) = 0$ can be quite small if highly accurate conformation is approximated by the simulation, the protein-molecule relative term $D\left(c_m + \delta, c_p\right) - D\left(c_m, c_p\right)$ have to be approximated by an additional molecule docking process. Even if the RDkit simulation does not provide an accurate conformation, the deviation is still larger when using the Single-Tower model. Both the effect of optimal rotation, translation, and conformation changes regarding the ligand torsion angles will lead to deviation. However, for the DrugCLIP model, only the torsion angles in ligand conformation are taken into account. As a result, the supervised-learning based methods have to rely on molecule docking software to get the optimal rotation $R$ and translation $t$.  $\qquad\square$

This mathematical derivation proves that our framework is more robust and will enjoy the advantages of introducing large amounts of noisy data for training.

# B    Implementation details

## B.1    Implementation of HomoAug

We propose a novel method called Homo-Aug, which utilizes the concept of homologous proteins in biology for data augmentation. Our core idea is to combine ligands from the PDBbind database with homologous proteins corresponding to their protein pockets, thereby generating new training data. Homologous sequences play a fundamental role in the domain of proteins, representing proteins that share a common ancestry in terms of evolutionary relationships. These homologous proteins exhibit certain resemblances in terms of their sequence, structure, and interactions with ligands. By incorporating homologous proteins alongside ligands, we introduce the noise of protein evolution, which can augment data while mitigates the risk of significant alterations in the binding properties of proteins and ligands.For our study, we opted to utilize the AlphaFold protein structure database [17, 42] as our search library for homologous proteins. This database leverages the AlphaFold2 [17] algorithm, enabling the prediction of protein structures for those lacking structural information but possessing

sequence data. To ensure the reliability and integrity of the database, we implemented a series of stringent filtering operations.Specifically, we retained only instances exhibiting high structural confidence, as indicated by residues with plDDT values exceeding 0.7 accounting for more than 90% of the protein structure. This filtering criterion ensured that our database comprised instances with robust structural predictions.Furthermore, to enhance the diversity of our database, we employed the MMseqs [12] algorithm to cluster the data using a 50% identity threshold. This clustering process remove the very similar protein , promoting greater variation within the database.Through these rigorous filtering and clustering operations, we obtained a comprehensive homologous retrieval database comprising 8,449,772 protein sequences, each paired with its corresponding reliable protein structure. Utilizing the provided database, we have expanded and enriched the instances sourced from the PDBBind database. Our approach involved several steps to ensure the quality and diversity of the data. Initially, instances containing non-standard residues or pockets with multiple chains were excluded from the dataset. This step was undertaken due to the inherent difficulty in searching for homologous protein complexes. Next, for each protein's pocket-containing chain, we employed the Jackhmmer [15] Algorithm to conduct a search for homologous proteins. The top 200 homologous proteins identified in the Jackhmmer search results were retained for each instance, thereby augmenting the dataset and enhancing its diversity.To ensure ligand binding within the pocket of the homologous protein, we performed structure alignment between the homologous proteins and the original proteins using the TMalign [50] algorithm. This alignment process aimed to identify similarities between the overall protein structure and the pocket region. In order to ensure the quality of the newly generated protein-ligand pairs, we retained only those that exhibited a sufficient degree of structural similarity. Specifically, we imposed the condition that the TMscore should be equal to or greater than 0.4, indicating a significant structural similarity, and the alignment rate of the pocket region should be equal to or greater than 40%, denoting a substantial alignment of residues within the pocket region.Finally, we extracted the atoms of the homologous proteins located within a 6Å radius of the ligand, defining this extracted region as the new pocket. This step allowed us to precisely delineate the pocket for ligand binding and subsequent analysis.

By employing the data augmentation method described earlier, we have achieved significant success in obtaining 758,107 novel pocket-ligand pairs. This approach has resulted in the expansion of 51% of the original instances sourced from the PDBbind database. The implementation of the Homo-Aug method allows us to effectively harness the concept of homologous proteins and utilize it to augment our training data. Through a comprehensive set of filtering and alignment operations, we have successfully enhanced the diversity of the data. This augmentation process significantly broadens the foundation for the field of drug virtual screening, offering a more comprehensive and varied dataset for subsequent analyses and investigations.

## B.2    Implementation of Fine-grained Atom Interaction

Besides aligning the representations of the global features from entire pockets and molecules, we also explore the usage of fine-grained features in our contrastive learning framework. When pretraining the 3D encoder, we also take the interactions between atoms into account. Specifically, we found out that in the complex structure, one single protein atom is only able to form strong interactions with a limited number of atoms from the binding molecule, and vice versa. From this biological intuition, we are able to propose an additional loss term that makes use of the fine-grained representation.

To define our training objective, we denote the atom-level representation of a molecule $i$ as $[m_i^1, m_i^2, \cdots, m_i^N]$ and the atom-level representation of a pocket $j$ as $[p_j^1, p_j^2, \cdots, p_j^M]$. To measure the alignment between the representations, we first employ a similarity metric as cosine similarity. Given an embedding $m_i^u$ in $m_i$, we compute its similarity with all tokens in $p_j$ and select the top K most similar tokens based on the similarity scores. We denote the set of indices of the selected tokens in $p_j$ as $\mathbf{T}_{p_j}$.

Similarly, for each token embedding $p_j^v$, we find its K most similar tokens in $m_i$ and represent the corresponding set of indices as $\mathbf{T}_{m_i}$.

Next, we defined the loss term as follows:

$$\mathcal{L}_{\text{topk-topk}} = \sum_{v \in \mathbf{T}_{m_i}} \sum_{u \in \mathbf{T}_{p_j}} \mathrm{s}\left(m_i^u, p_j^v\right) \tag{11}$$

By optimizing this topk-topk loss term, we encourage the model to focus on the most informative atom alignments, facilitating better representation on the fine-grained level. When implemented we add the topk-topk loss term as an auxiliary loss to the global-level contrastive learning objective as in Eq. 5. We also conduct experiments by extracting atom-level representations from different layers of the encoder to compare the difference. The experiment result for atom-level interaction is shown in section C.2.

## B.3 Evaluation Metrics

There are several evaluation metrics we use in this paper for benchmarking virtual screening tasks. Here are the detailed explanations.

**BEDROC** incorporates exponential weights that assign greater importance to early rankings. In the context of virtual screening, the commonly used variant is $\text{BEDROC}_{85}$, where the top 2% of ranked candidates contribute to 80% of the BEDROC score (cite). The formal definition is:

$$\text{BEDROC}_\alpha = \frac{\sum_{i=1}^{\text{NTB}_t} e^{-\alpha r_i/N}}{R_\alpha \left(\frac{1-e^{-\alpha}}{e^{\alpha/N}-1}\right)} \times \frac{R_\alpha \sinh(\alpha/2)}{\cosh(\alpha/2) - \cosh(\alpha/2 - \alpha R_\alpha)} + \frac{1}{1 - e^{\alpha(1-R_\alpha)}}. \tag{12}$$

**Enrichment Factor(EF)** is also a widely used metric, which is calculated as

$$\text{EF}_\alpha = \frac{\text{NTB}_\alpha}{\text{NTB}_t \times \alpha}, \tag{13}$$

where $\text{NTB}_\alpha$ is the number of true binders in the top $\alpha\%$ and $\text{NTB}_t$ is the total number of binders in the entire screening pool.

We also adopted **ROC enrichment metric (RE)**, which is calculated as a ratio of the true positive rate to the false positive rate (FPR) at a given FPR threshold:

$$\text{RE}(x\%) = \frac{\text{TP} \times n}{\text{P} \times \text{FP}_{x\%}}, \tag{14}$$

where $n$ is the total number of compounds, TP is the number of compounds that are correctly identified as active, P is the total number of active compounds, and $\text{FP}_{x\%}$ is the number of false positives predicted at a specified rate (e.g. 0.5%, 1%, etc.).

## B.4 Encoder Pre-training

Our pre-training of the molecule and pocket encoders is based on the methodology proposed by UniMol [55]. Similar to BERT [5], we utilize a masked token prediction task. In the context of molecule or pocket data, this task involves predicting masked atom types. To augment the complexity of the pre-training task and extract valuable insights from 3D coordinates, we introduce an additional task called position denoising. Specifically, we add random uniform noise within the range of $[-1\text{Å}, 1\text{Å}]$ to 15% of the atom coordinates. Two tasks are incorporated to restore the original positions. Firstly, the model needs to predict the original distance between two corrupted atoms. Secondly, the model needs to estimate the original coordinates of a corrupted atom using the SE(3)-Equivariance coordinate system.

## B.5 Contrastive Learning Training Details

We train our model using the Adam optimizer with a learning rate of 0.001. The other hyper-parameters are set to their default values. We have a batch size of 192, and we use 4 NVIDIA A100 GPU cards for acceleration. We train our model for a maximum of 200 epochs. To avoid overfitting, we use the CASF-2016 dataset as a validation set and select the epoch checkpoint with the best $\text{BEDROC}_{85}$. For more detailed training configurations, please refer to the code.

For the model used for human evaluation(DrugCLIP-L), we use dot product as the distance metric. For other models we use cosine similarity.

# C Additional Experiments

## C.1 Evaluation on Target Fishing

Since DrugCLIP has the ability to learn the matching between proteins and molecules, it could be also used for target fishing, another important task in drug discovery, which entails the identification of the target from a pool of candidate targets that have the potential to bind to a specific molecule. We establish a benchmark using the CASF-2016 dataset. For each molecule, we test whether the model can correctly find its corresponding pocket from all other pockets. As shown in Table 6, DrugCLIP exhibits superior accuracy in the top 1 to 5 predictions as compared to docking software, i.e. Glide, and Vina. Conversely, DrugBA performs much poorer, with results comparable to random guessing.

Note: In this benchmark, we are unable to use the CASF-2016 dataset as both the test set and the validation set. Therefore, we split our training set in a 9 to 1 ratio and allocate the latter portion as the validation set.

Table 6: Result of Target Fishing Task on CASF-2016 dataset

| | Accuracy | | | | |
|---|---|---|---|---|---|
| | @1 | @2 | @3 | @4 | @5 |
| Vina [41] | 3.38 | 5.26 | 7.52 | 9.02 | 10.15 |
| Glide [11] | 14.98 | 22.85 | 30.34 | 35.58 | 39.33 |
| DrugBA | 0.37 | 0.74 | 1.11 | 2.22 | 2.22 |
| DrugCLIP | 24.07 | 42.96 | 51.11 | 59.26 | 62.59 |

## C.2    Global and Local interactions

Table 7: Performance Comparison on DUD-E and LIT-PCBA Datasets by adding atom-level interactions

| | DUD-E | | |
|---|---|---|---|
| | AUROC % | BEDROC % | EF@1% |
| Global only | 80.93 | 50.52 | 31.89 |
| with last | 78.87 | 44.72 | 28.65 |
| with second | 82.79 | 50.57 | 32.45 |

As shown n Table 7, using atom embeddings from the last transformer layer yields worse performance. However, marginal improvement is observed when utilizing embeddings from the second last layer. Selecting the appropriate transformer layer is crucial for obtaining effective atom embeddings and enhancing model performance, and should be considered as future work.

Table 8: Results of Human Expert evaluation. During the experiments, experts are free to choose their preferable computational tools to evaluate the screened molecules.

| | 5kdt | 6g2o | 1n5x | 7ksi | 8etr |
|---|---|---|---|---|---|
| Glide [11] | 2 | 2 | 4 | **7** | 4 |
| DrugCLIP | **8** | **8** | **6** | 3 | **6** |

## C.3    Evaluation on Kinase Screening

In order to evaluate whether our data augmentation has negative effects on screening for similar protein pockets, we evaluate our methods on a carefully curated kinase dataset. we build a novel dataset that provides protein structures of the kinase and the molecule structures of the inhibitors. To assemble this dataset, we meticulously collected 154 kinase structures sourced from the KLIFS database (https://klifs.net/). Complementing this, we harnessed the power of data mining techniques to derive a collection of 9423 inhibitor structures from the ChEMBL database with reported bioactivity data. We further took the subset by intersecting our dataset with FDA-approved drugs, as the protein binding of drugs has been extensively explored by researchers. In our experiments, the goal is to identify the correct kinase within the top-ranked pocket structures for the given inhibitor structure input.

It's noteworthy that DrugCLIP surpasses the performance of both the molecular docking method (AutoDock Vina) and the Unimol binding affinity prediction model. We selected Unimol as a representative due to its being open-sourced and its achievement of state-of-the-art excellence in various ligand-pocket tasks, including the affinity prediction task, outperforming alternatives such as Planet.

## C.4    GPCR

In this section, we demonstrate the ability of our model to pair all known human GPCR proteins with 31,422 human metabolites using AlphaFold2 predicted models. We aim to identify unrevealed GPCR ligands to facilitate functional studies, as certain GPCR proteins may have unexpected functions. For example, hOF17-4, an olfactory receptor, locates on sperms and contributes to egg localization. To achieve this, we utilized Fpocket

Table 9: Results for kinase screening

| Method | Top5 acc(%) | Top10 acc(%) | Top15 acc(%) | Top20 acc(%) | Time |
|---|---|---|---|---|---|
| Vina | 8.51 | 14.89 | 24.47 | 34.04 | 4h10min |
| Unimol Regression | 7.45 | 18.09 | 23.40 | 32.98 | <1min |
| DrugCLIP | **14.89** | **23.40** | **28.72** | **37.23** | <1min |

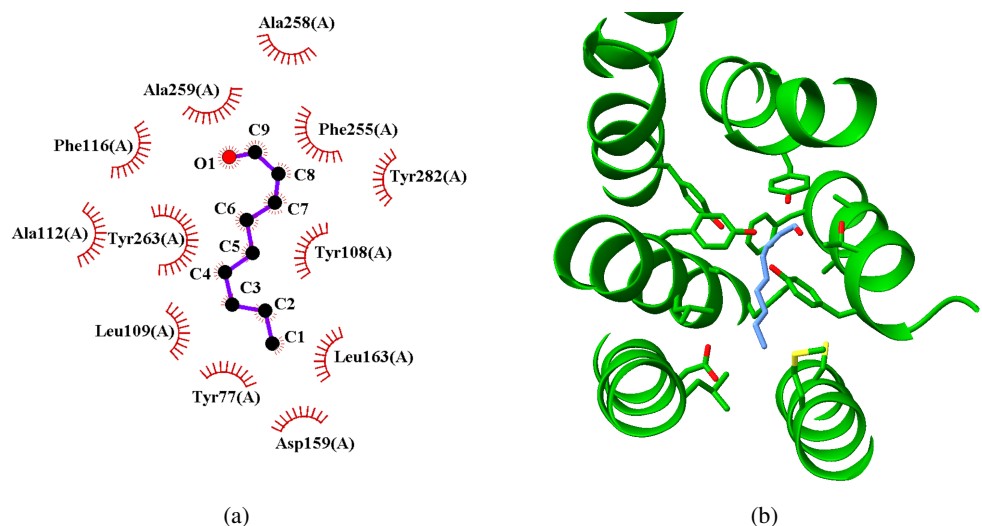

(a)          (b)

Figure 6: Visualization of the docking pose of OR2T5 and 2-nonenal complex. The 2D interaction pattern is generated with LigPlot+. Interactions between OR2T5 and 2-nonenal are mainly hydrophobic interactions.

for ligand-binding pocket detection on GPCR protein surfaces and obtained 17,702 pockets. Evaluating more than $5 \times 10^8$ pocket-ligand pairs would typically take around one CPU year with cutting-edge active-learning-assisted docking; however, our model can rank these pairs within minutes.

We manually evaluated top-ranked pairs and predicted their binding poses using commercialized docking software GLIDE in the Schrodinger Suite. Our findings revealed several particularly interesting pairs, including three kidney-enriched olfactory GPCRs, OR2T5, OR2T11, and OR4C3, which were predicted to bind known metabolic wastes. The kidney-expressed olfactory system has long been known to influence urine production. Additionally, the presence of olfactory G protein, $G_{olf}$, and olfactory-related adenylate cyclase AC3 was detected in the distal convoluted tubule. When olfactory signaling was blocked via AC3 knock-out, creatinines accumulated in the blood, indicating defective renal function.

Our model identified OR2T5 paired with 2-nonenal, OR2T11 paired with p-cresol, and OR4C3 paired with D-lactic acid. Docking poses revealed potential hydrophobic interactions, hydrogen bonds, and $\pi - \pi$ interactions between pockets and ligands. As previous studies reported, 2-nonenal is a uremic toxin; p-cresol is an intermediate of tyrosine metabolism; and D-lactic acid is a widely distributed waste product. These molecules are highly toxic and require timely cleaning/recycling by either the excretory system or cellular processes. Our findings suggest that olfactory receptors in the kidney can sense metabolic wastes and regulate the excretion process as a feedback loop. Visualizations are shown in Figure 6,7,8.

## D   Disscussion on the Impact of Molecule Library

Besides library size, the molecule library choice also influences the process of identifying true binders. However, discussions on expanding the chemical library and its relation to finding potential drugs have been ongoing. Previous work [25] compared billion-size and million-size libraries, showing a log-linear improvement in predicted affinity scores with library size, and reduced bias towards reported drugs in larger libraries. Other research [19] also found that using an ultra-large library enhances the quality of screened hits compared to using small libraries which might have been fully reported by patents and articles. These studies indicate that ultra-large libraries not only offer improved hit quality due to their scale but also due to their reduced bias and

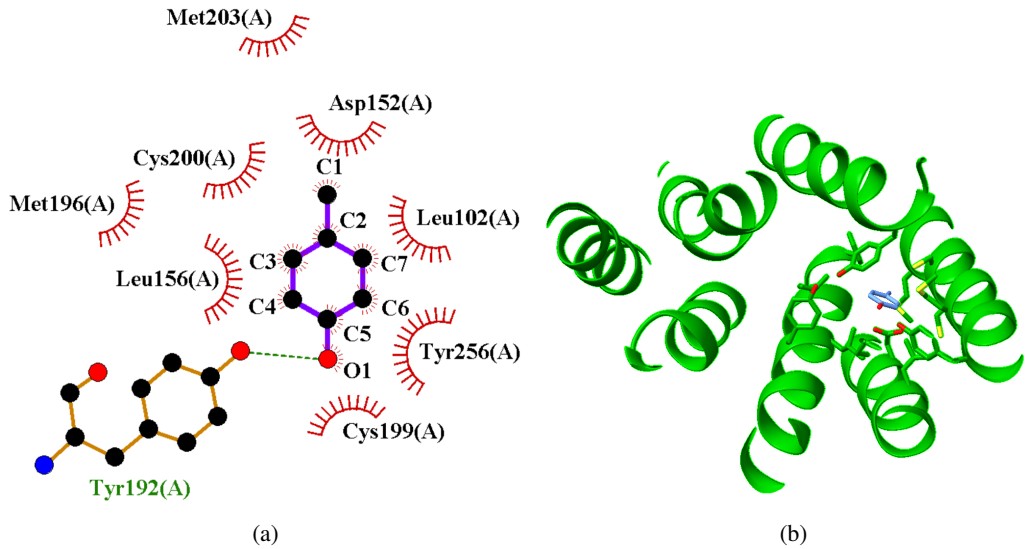

(a)                                            (b)

Figure 7: Visualization of the docking pose of OR2T11 and p-cresol complex. The 2D interaction pattern is generated with LigPlot+. Tyr192 of OR2T11 and O1 of p-cresol form a hydrogen bond. Tyr256 could have potential $\pi - \pi$ interaction with p-cresol.

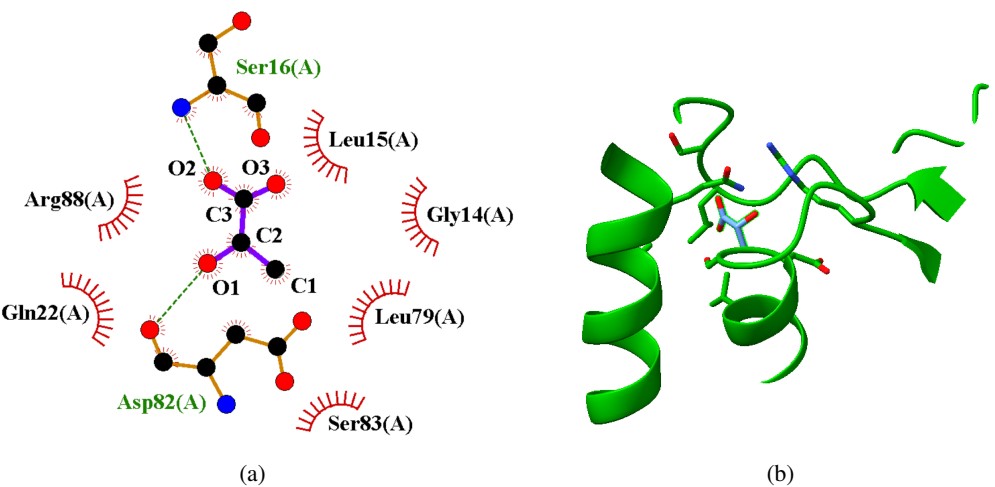

(a)                                            (b)

Figure 8: Visualization of the docking pose of OR4C3 and D-lactic acid complex. The 2D interaction pattern is generated with LigPlot+. Ser16 and Asp82 interact with D-lactic acid via hydrogen bonds.

lesser prior exploration, in contrast to smaller libraries. As a conclusion, an ultra-fast virtual screening is of great importance, because it offers the opportunity to utilize those ultra-large libraries.

## E    Limitations

The major limitation of our paper lies pertains to its interpretability. Although our model demonstrates enhanced effectiveness and efficiency, it falls short in terms of interpretability compared to traditional docking methods. These conventional approaches offer visualizations that elucidate the binding mechanism between a pocket and a molecule, providing clear explanations.

## F    Negative Societal Impacts

While our method has the potential to greatly expedite the drug discovery process, which is undoubtedly advantageous, it is important to consider the potential implications it may have on drug auditing and clinic trials. The increased speed and efficiency may inadvertently create additional pressures and challenges for regulatory bodies responsible for ensuring the safety and efficacy of new drugs.

