# OpenReview forum: "DrugCLIP: Contrastive Protein-Molecule Representation Learning for Virtual Screening"
_NeurIPS.cc/2023/Conference — NeurIPS 2023 poster_

### Official Review · Reviewer_GQNT · 2023-06-29

**Soundness:** 3 good
**Presentation:** 3 good
**Contribution:** 3 good
**Rating:** 7
**Confidence:** 4

**Summary:**

This paper proposes a new virtual screening framework DrugCLIP, which aims to identify molecules most likely to bind to a target.
Inspired by the recent multi-modal learning approach CLIP, the authors propose to utilize contrastive learning for learning representations of molecules and proteins.
Authors reformulate the drug binding problem based on the representations into an information retrieval task.

**Strengths:**

- By adopting a self-supervised multi-modal learning approach to binding prediction problems, DrugCLIP outperforms previous work that utilizes labels which is costly in time and finance.
- Augmentation technique inspired from domain knowledge is proposed, which is novel.
- Extensive experiments have been conducted, including human evaluation.


**Weaknesses:**

- More detailed explanation of the experimental setting will be helpful for the readers.
- In line 109, the authors mentioned "an end-to-end manner," which may mislead the readers that the whole framework is. It should be clarified more since the DrugCLIP is a two-phased method.
- In biological data, many missing data exist, which should not be considered a negative sample, as done in section 3.3. Recent self-supervised learning approaches deal with such false negatives that can also be adopted in this framework.

**Questions:**

In my knowledge, the inherent principle in self-supervised learning is that "multiple views of the image should be consistent in representation space." This principle can also be applied to multi-modal learning: "representation of text and images that share the same semantic should be consistent in representation space."

But I'm not sure why DrugCLIP works in the sense that binding molecules and proteins should be closer for binding. Do you have any intuitive explanation for that?

**Limitations:**

Provided in Weakness section.

---

> ### Author Rebuttal · Authors · 2023-08-08
>
> ## Response to Reviewer GQNT
>
> We sincerely thank the reviewer for the positive feedback. Your support and encouragement are greatly appreciated. We have addressed all your concerns in the following responses.
>
> ### Q1: Detailed explanation of the experimental setting
>
> We apologize for any inconvenience that may have been caused while reading our original paper. Most of the experimental settings are provided in the Appendix to save space. We will list some of the key experimental settings here.
>
> For the virtual screening experiments, we mainly evaluate our model on DUD-E and LIT-PCBA. The evaluation metrics we applied are AUROC, BEDROC, and EF.
>
> We trained our model on PDBBind and ensured that proteins with the same pdbid were excluded from the test set. As for the baseline models, all reported metrics were obtained directly from the original papers. We used the ADAM optimizer, with the learning rate equals to 0.001. We trained our model on 4 A100 GPUS with a batch size of 4x48. We select the ckpt with best valid BEDROC and the patience is set to 20.
>
> Regarding the Efficiency Analysis, we conducted tests on our methods, docking, and other baseline methods using a single A100 GPU and CPU with 128 cores. Due to the extensive time required to run molecular docking on the full Enamine dataset, we opted to perform the docking on a smaller subset. Using the average docking time from this subset, we estimated the resource usage on the entire dataset.
>
> ### Q2: Improper notion of "an end-to-end manner"
>
> For sure, we adopt a two-phase procedure for training the DrugCLIP model. We will revise our paper and remove the notion of "an end-to-end manner" for better presentation. Thanks for pointing it out!
>
> ### Q3: Many missing data require sampling technique to reduce false negatives
>
> We sincerely thank you for offering us this valuable insight. We totally agree that simply adopting in-batch negative sampling might raise problems when there are a lot of missing data. However in this filed, only an extremely small proportion of molecules exhibit binding affinity towards the target protein in the vast chemical space. Consequently, the likelihood of sampling false negative data remains comparatively low. Besides, it is essential to note that contrastive learning diverges from supervised learning methods that rely on binding affinity labels.
>
> Contrastive learning possesses the capability to discern the relationship between positive and negative examples by leveraging the condition that the majority of negative instances exhibit inferiority when compared to the positive ones. Hence, contrastive learning exhibits a greater capacity for noise tolerance as compared to directly regressing to a specific metric as in supervised learning.
>
> To the best of our knowledge, certain prior studies have employed molecule similarity to filter out similar structures. However, this approach may not be suitable when considering the cliff effect. In our future research, we aim to explore sampling techniques inspired by biological domain knowledge to address this limitation and enhance the accuracy of our methodology.
>
> ### Q4: Intuitive explanation for binding molecules and proteins should have close representations
>
> There is a compelling rationale behind this statement, drawing from both biological and machine-learning perspectives.
>
> In the realm of drug discovery, widely known aphorisms, such as 'similar molecules bind to the same pocket' and 'similar pockets bind to the same molecules,' have significantly influenced drug design [1,2]. Inspired by these intuitions, it is reasonable to project molecules and pockets into a shared latent space and align their representations from binding pairs. Upon observing pocket-ligand structures at a more intricate level, we often find numerous mutually corresponding features, such as H-Bond donors/acceptors and positive/negative charges. These features substantiate the notion that binding molecules and proteins should exhibit close representations, underpinning our approach with strong biological knowledge support.
>
> From an alternative perspective, the notion of binding molecules and proteins can be likened to the relation between user and item in recommender systems, whose objective is to recommend appropriate items to users. The user-item relationship mirrors the Target-molecule association. Similar to the aforementioned aphorisms, similar users may like the same items, and similar items may pique the interest of the same user. Throughout the evolution of recommender systems, collaborative filtering and latent factor models have been widely used to quantify the relation between user and item, employing a similarity function based on latent representations [3]. It is evident that our conjecture, that binding molecules and proteins should possess closely aligned representations, resonates quite strongly.
>
> [1] Hoffmann et al, A new protein binding pocket similarity measure based on comparison of clouds of atoms in 3D: application to ligand prediction, 2010, BMC bioinformatics
> [2] Chackalamannil et al, Comprehensive medicinal chemistry III, 2017, Elsevier
> [3] Koren et al, Matrix Factorization Techniques for Recommender Systems, 2009, Computer

---

> > ### Comment · Reviewer_GQNT · 2023-08-15
> > **Thank you for author rebuttal**
> >
> > We sincerely thank the authors for their effort in addressing my concerns. Also, domain knowledge on **why contrastive learning on molecules and proteins** would be helpful for readers in understanding the effectiveness of contrastive learning in molecule-protein representation learning.
> >
> > Therefore, I will raise my score to 7.

---

### Official Review · Reviewer_cZpn · 2023-07-03

**Soundness:** 4 excellent
**Presentation:** 3 good
**Contribution:** 4 excellent
**Rating:** 8
**Confidence:** 5

**Summary:**

The authors recast virtual screening as an information retrieval problem: by learning appropriate representations of both proteins and molecules, and taking contrastive loss based on binding affinity between protein-molecule pairs, the aim is to learn a model where protein queries passed through one encoder can be used to retrieve a molecule from a large-scale library. The design of the model allows training data from a range of sources such as PDBBind, but also ChEMBL and BioLip, as well as augmentation with protein structures based on protein homology across evolution.

The method is benchmarked using DUD-E and LIT-PCBA, and by human evaluation of comparative examples from Glide, a commercial docking system. The method is demonstrated to outperform most finetuned deep learning methods on the benchmarks and produce better binding molecule sets than Glide in 80% of cases, as judged by human experts.

**Strengths:**

The authors suggest a good method for obtaining aligned representations for proteins and molecules, where the alignment is induced by protein binding affinity, using a contrastive loss. The representations of molecules are undertaken in 3D which is still not the default in this field, despite its clear importance in protein-affinity, and uses a biologically-plausible data augmentation.

The results in the DUD-E benchmark are strong, particularly in a zero-shot setting, significantly out-performing other methods. Results on LIT-PCBA are also strong, with enrichment factors far above the performance achieved with commercial docking software. The results obtained on the time taken to virtually screen large libraries are impressive -- there are clear advantages over commercial docking software, and machine-learned scoring functions.

In addition, the clear ablation studies and human evaluation add nice details to the work. Overall this work is a very strong addition to the techniques available for screening and it will be exciting to see it in use.

**Weaknesses:**

There are no significant weaknesses in this paper.

**Questions:**

No questions

**Limitations:**

No limitations

---

> ### Author Rebuttal · Authors · 2023-08-08
>
> ## Response to Reviewer cZpn
>
> We extend our heartfelt gratitude to the esteemed reviewer for their generous appraisal of our work and for awarding high scores. Your positive feedback and acknowledgment of our efforts are deeply appreciated. We are thrilled that our research has met your expectations, and we sincerely thank you for taking the time to review our submission. Your encouragement motivates us to continue striving for excellence in our endeavors.

---

> ### Comment · Area_Chair_ZAJ8 · 2023-08-16
> **Respond to authors' rebuttal**
>
> Please, look at the authors' rebuttal and the other reviewers' comments and indicate if you would like to change anything in your review.

---

### Official Review · Reviewer_8ynk · 2023-07-04

**Soundness:** 3 good
**Presentation:** 3 good
**Contribution:** 2 fair
**Rating:** 4
**Confidence:** 4

**Summary:**

The authors proposed a contrastive learning framework, DrugCLIP, for the drug virtual screening task which identifies potential drugs from vast compound databases to bind with a particular protein pocket. It reformulates virtual screening as a dense retrieval task and employs contrastive learning to align representations of binding protein pockets and molecules from a large quantity of pairwise data without explicit binding-affinity scores. Specifically, the framework computes a contrastive loss between two separate pre-trained encoders to maximize the similarity between a protein-molecule pair which can binding together and minimize it otherwise. Besides, the authors introduce a biological-knowledge inspired augmentation method, HomoAug, which creates protein-molecule pairs based on protein homology evolutions. Experiments on two challenging virtual screening benchmarks, demonstrate that zero-shot performance of this model surpasses most deep learning baselines that carefully finetune on labeled data.

**Strengths:**

1.	The proposed DrugCLIP framework reformulates virtual screening as a dense retrieval task and employs contrastive learning to align representations of binding protein pockets and molecules from a large quantity of pairwise data, which provides researchers a new perspective for virtual screening.
2.	The designed contrastive loss relieve the dependency on explicit labeling of binding affinity, and facilitates the usage of large-scale unlabeled data beyond densely annotated small datasets (such as PDBBind).
3.	The dense retrieval characteristic of DrugCLIP brings high efficiency to online inference and promising high-throughput virtual screening on billions of molecules.
4.	A biological-knowledge inspired augmentation method named HomoAug are proposed, which creates protein-molecule pairs based on protein homology evolutions. The shortage of public data is indeed obvious in drug discovery, and this augmentation method may help alleviate it.
5.	The organization of this paper is very clear and easy to understand.


**Weaknesses:**

1.	The main contribution of this paper is to directly apply the contrastive learning CLIP to the virtual screening scenario, which is maybe insufficient to support a poster published in NeurIPS.
2.	The proof of Proposition 1 seems insufficient to support training-test consistency. Firstly, the model ability of f_theta and k_theta is ignored. The docking methods (k_theta here) can directly capture protein-ligand interactions and maybe more natural to adapt to conformation perturbations. Secondly, the binding conformations of ligands are different from their free states probably, and thus the optimal Rotation R and translation t cannot be found. Thirdly, many SOTA docking methods employ two-tower frameworks, such as Equibind, TANKBind, DiffDock, etc.
3.	The model lacks further experiments on the screening power compared to docking methods in CASF-2016 which is an important benchmark for in silico drug discovery.
4.	Lack of interpretability. Although this model demonstrates enhanced effectiveness and efficiency, it falls short in terms of interpretability compared to docking methods, as the authors mentioned in the appendix. These conventional approaches offer visualizations that elucidate the binding mechanism between a pocket and a molecule, which is intuitive and reliable for chemistry researchers to check the rationality of receptor-ligand interactions and modify the molecule structures.


**Questions:**

The questions are basically mentioned in the “weaknesses” part above.
1.	The proof of Proposition 1 seems insufficient to support training-test consistency. Firstly, the model ability of f_theta and k_theta is ignored. The docking methods (k_theta here) can directly capture protein-ligand interactions and maybe more natural to adapt to conformation perturbations. Secondly, the binding conformations of ligands are different from their free states probably, and thus the optimal Rotation R and translation t cannot be found. Thirdly, many SOTA docking methods employ two-tower frameworks, such as Equibind, TANKBind, DiffDock, etc.
2.	The model lacks further experiments on the screening power compared to docking methods in CASF-2016 which is an important benchmark for in silico drug discovery.
3.	Lack of interpretability. Although this model demonstrates enhanced effectiveness and efficiency, it falls short in terms of interpretability compared to docking methods, as the authors mentioned in the appendix. These conventional approaches offer visualizations that elucidate the binding mechanism between a pocket and a molecule, which is intuitive and reliable for chemistry researchers to check the rationality of receptor-ligand interactions and modify the molecule structures. Try to give some correlation analysis and demonstrate some interaction patterns learned by DrugCLIP and not just show the performance. That would be more convincing for biochemistry scientists.


**Limitations:**

Lack of interpretability. Although this model demonstrates enhanced effectiveness and efficiency, it falls short in terms of interpretability compared to docking methods, as the authors mentioned in the appendix.

---

> ### Author Rebuttal · Authors · 2023-08-08
>
> ## Response to Reviewer 8ynk
>
> We greatly appreciate your valuable feedback. We are committed to refining our work to enhance its quality and impact.
>
> ### Q1: Regarding our paper's contribution
>
> Our paper's main contribution lies not merely in applying contrastive learning, specifically CLIP, to the virtual screening (VS) task and obtaining commendable results. Instead, our focus is on pioneering a new information retrieval paradigm for approaching tasks within the VS domain instead of the existing regression or classification views. This innovative method enables searching through billion-scale molecular libraries in several minutes while maintaining a high recall rate.
>
> Upon transforming virtual screening (VS) into a similarity matching problem between proteins and molecules, it is crucial to highlight the non-trivial nature of the contrastive learning concept in this scenario. In the original context, images, and text serve as parallel mediums, each offering distinct perspectives about a shared subject. Therefore, it is natural for CLIP to utilize contrastive learning to learn this joint representation by differentiating similar ones from dissimilar ones. In contrast, the relationship between a drug and its target is non-parallel, highlighting a deep-seated binding relationship that underscores their intricate interplay. Given this, the core principles of contrastive learning as applied in the drug-target domain differ markedly from those in the text-image domain. Inspired by collaborative filtering and the latent factor model in the recommender system that the relationship between user and item could be formulated as a similarity function based on their latent representations, we propose to view protein and molecule analogically and aim to learn their latent representation by distinguishing positive and negative pairs in a contrastive learning manner. However, limited true negative binding protein and molecule pairs are provided in the field of virtual screening. Fortunately, if a specific protein-molecule pair has already been verified to exhibit a binding relationship, it is probable that they possess a negative binding relationship with other molecules/proteins. This observation can be incorporated as an in-batch sampling strategy. It is noteworthy that while the techniques employed in DrugCLIP bear resemblance to standard CLIP methods, the underlying motivation and principles differ significantly.
>
> In terms of our other technical contributions, our paper introduces HomoAug, an innovative pocket augmentation method. The significance lies in its application to biological data, where data augmentation presents intricate challenges. We also explore advanced modeling techniques to meticulously capture the nuanced atomic interactions between the ligand and the pocket. You'll find a comprehensive exploration of these findings within the Appendix. While these contributions are noteworthy, they do not form the central focus of our paper. Therefore, we've deliberately chosen to reserve a detailed discussion of these aspects for the Appendix, indicating their potential as promising avenues for future research.
>
> ### Q2: The proof of Proposition 1 seems insufficient
>
> Thank you for your feedback. Proposition 1 aims to show the superior robustness of the two-tower architecture employed in DrugCLIP compared to single-tower models when provided with inaccurate 3D structures as input. The inputs for both models are molecule conformations that have not been docked into the protein.
>
> The key rationale behind the proof lies in the fact that deviation of the two-tower model is not reliant on inter-distances. We apologize for any confusion regarding the optimal R and t values; this term may not be as small as initially indicated. Nevertheless, the error of the single-tower model remains higher due to additional inter-distance deviation. Consequently, the single-tower model is more reliant on docking accuracy, whereas the dual-tower model can accommodate unbound structures as input.
>
> Indeed, as you have indicated, there exist docking models that can be employed for screening. However, it should be emphasized that a prior study[1] showcased the inferior performance of blind docking approaches, such as Equibind, TANKBind, and Diffdock, when transitioning from blind docking to local docking. While these methods excelled in locating the pocket, they are not suitable for screening purposes.
>
> [1] Yu et al, Do Deep Learning Models Really Outperform Traditional Approaches in Molecular Docking?,2023,arxiv
> ### Q3: Lack important benchmark CASF-2016
>
> Thanks for providing a valuable benchmark. We assessed DrugCLIP and molecular docking methods using the CASF-2016 screening and target fishing tasks. The performances of the target fishing task are already in our Appendix, while the screening results are presented in the following Table.
>
> | Method | top 1 | top 2 | top 3 | top 4 | top 5 |
> |--------|-------|-------|-------|-------|-------|
> | vina   | 0.034 | 0.049 | 0.064 | 0.083 | 0.109 |
> | glide  | 0.195 | 0.270 | 0.360 | 0.386 | 0.416 |
> | Ours   | 0.259 | 0.411 | 0.500 | 0.581 | 0.637 |
>
> It is pretty obvious that our method outperforms molecular docking by a large margin. We will include the above table in our revised paper.
>
> ### Q4: Lack of interpretability compared to docking
>
> Thanks for your interest in the interpretability of our work. Predicting affinity with binding poses is crucial for drug discovery. We aim to augment, not replace, traditional docking, enabling larger chemical library screening. DrugCLIP suits a multi-step workflow, with subsequent docking or MD simulation for binding poses.
>
> Considering the challenge of docking a billion-size library, our approach streamlines by selecting the top 1 million. Docking them is feasible, expanding the search from 1 million to a billion. DrugCLIP excels in multiple benchmarks, offering an edge in exploring a broader, accurate range of drug candidates.

---

> ### Comment · Area_Chair_ZAJ8 · 2023-08-16
> **Respond to authors' rebuttal**
>
> Please, look at the authors' rebuttal and the other reviewers' comments and indicate if you would like to change anything in your review.

---

> > ### Comment · Area_Chair_ZAJ8 · 2023-08-19
> > **Reminder**
> >
> > A reminder of this.

---

### Official Review · Reviewer_Bb5s · 2023-07-06

**Soundness:** 2 fair
**Presentation:** 2 fair
**Contribution:** 2 fair
**Rating:** 4
**Confidence:** 5

**Summary:**

## DrugCLIP: Contrastive Protein-Molecule Representation Learning for Virtual Screening

The authors present DrugCLIP, a contrastive learning method for virtual screening. DrugCLIP is designed to align representations of small molecules and protein binding pockets. By using a contrastive learning framework, the authors are able to use data without associated affinity measurements, which are relatively scarce. Additionally, the authors present a data augmentation technique based on identifying homologous proteins.


**Strengths:**

The authors employ an interesting framing of virtual screening and achieve good performance across a small panel of benchmark datasets with improved speed compared to more expensive traditional methods. The authors compare their method to several existing approaches and perform additional analysis around the practical usability and behviour of the model.

**Weaknesses:**

* Clarity and explanatory detail could be improved throughout
* The data augmentation method seems dubious. Whilst it clearly provides more data, selectivity is obviously a vital aspect of developing therapeutics. By using these augmentations it does not seem to this reviewer that the ability of the model to determine selective binding is well-served and really serves to smooth what is fundamentally not a smooth function due to cliff effects.
* It would be interesting to explore performance on benchmarks of highly related targets and ligands, such as the KIBA and Davis Kinase inhibitor datasets.


### Minor Comments
* Title: Typo in "Contrasive"
* This reviewer disagrees with the notion that the key issue is to identify which molecules bind to a particular target. While this is an important subproblem, it fails to address downstream problems. Co-crystallised structures and affinity measurements will still be required in experimental campaigns, even if we had a reliable binder identification method. These are the time-consuming and expensive steps. Docking and affinity prediction methods address both implicitly address the binder identification problem, whilst also providing a proxy for one of these expensive measurements. While docking is time consuming, it produces a richer output: both pose and score.
* L20-22 This is clearly true but perhaps lacks some nuance. Library composition must also have a significant impact.
* The authors state to the best of their knowledge this is the first information retrieval based framing of the virtual screening. I think this claim holds water but there does exist very similarly motivated prior work on a related application that should be discussed and referenced for completeness [1].
* L71 value judgments such as "Innovative" and "superb" (L73) should be avoided
* L127 the authors should be explicit about the type of noise added in the corruption processes.
* L197 it would provide better clarity if the authors could use technical terms such as apo and holo
* It would be helpful if the authors could be more descriptive about the noising procedure using RDKit. The authors refer to simulation but it is not immediately clear on what this is in reference to.


[1] https://www.biorxiv.org/content/10.1101/2022.04.26.489341v1.abstract


**Questions:**

* How are ligands combined with the pocket in the augmented dataset?*
* Do the authors perform any pre-processing of the structures, such as relaxation?
* I'm a little confused by the description of SE(3) Equivariance. The task is suited to invariance and the input appears to be invariant (based on discussion of distance-based encodings of structure).
* Masking atom types -> Is this masking the element type or the atomic identifier (e.g. the 37 standard atoms in Proteins)
* Do the authors apply any quality thresholding (e.g. based on pLDDT) for the data augmentation technique?
* I would appreciate if if the authors could expand on the purpose of adding the central node ([CLS])?
* Why do the authors filter out proteins with only one known pocket in the ChEMBL dataset?
* Why do the authors not report BEDROC on DUD-E Finetune?
* The authors state they remove all targets present in DUD-E from the training data. How exactly is this performed?
* For the human evaluation, it would be interesting to present docked structures to the experts as well as the molecular structures.

**Limitations:**

Technical weaknesses discussed above. No ethical concerns.

---

> ### Author Rebuttal · Authors · 2023-08-09
>
> ## Response to Reviewer Bb5s
>
> We sincerely thank the reviewer for your valuable advice. We will address the typos and word usage issues as you have pointed out. Regarding the questions about implementation details, we have included them in the global response.
>
> ### Q: L20-22: library composition also has a significant impact
>
> Agreed, library composition significantly impacts virtual screening. Recent research indicates that ultra-large libraries not only offer improved hit quality due to their scale but also due to their reduced bias and lesser prior exploration, in contrast to smaller libraries. In the original paper, we prioritized efficient virtual screening with a large library and didn't fully delve into the composition-size relationship. This will be discussed in the revised version.
>
> ### Q: There exists prior work using information retrieval for VS
>
> Thanks for pointing out this work that treats virtual screening as pocket matching. While it proposes pocket pretraining to find ligands for similar pockets, our approach is different. We directly align 3D representations of unbound molecule conformations and protein structures, without needing explicit pocket and ligand binding info. We'll cite this paper in our revised version and clarify the differences, emphasizing our unique contributions.
>
> ### Q: Lack of binding structures and affinity measurements compared to docking
>
> We agree that complex structures and affinity measurements are essential. Rather than replace conventional docking, DrugCLIP augments the process by allowing the screening of a larger chemical library. Our method is suitable for a multi-step workflow, with subsequent docking or MD simulation. Considering the challenge of docking billion-size libraries, our method streamlines selection, enabling researchers to conduct docking within a feasible timeframe. It expands the search space and outperforms other methods both in efficiency and accuracy.
>
> ### Q: Confusion of SE(3) Equivariance
>
> We apologize for not explaining SE(3) equivariance in our paper. Our model uses SE(3) equivariant EGNN heads for the pretraining task involving denoising coordinates.  However, in the screening task, we exclusively utilize invariant representations and do not employ the equivariant coordinates. We'll update our description to clarify this.
>
> ### Q: More experiments on the metric of BEDROC on DUD-E
>
> We didn't report the BEDROC results for the finetuning experiment because they were absent in the baselines, and the lack of code hindered our ability to obtain them. Nevertheless, our method attained a notable BEDROC result of 71.71 on the DUD-E dataset.
>
> ### Q: The implementation details of removing overlap targets
>
> In this paper, we adopted a straightforward approach to address the issue of overlapped protein targets in the training set by removing the corresponding pdbid, following other baselines in our comparison. We further explored using sequence identity at 90% thresholds to mitigate similar targets. The results are detailed in the following table.
>
> Table 1. Results on DUD-E with strict overlap-removing technique
> | Method | AUC   | BEDROC | EF @0.5% | EF @1% | EF @5% |
> |--------|-------|--------|----------|--------|--------|
> | Glide(docking)  | 76.7  | 40.7   | 19.39    | 16.18  | 7.23   |
> | Vina(docking)   | 71.6  | N/A    | 9.13     | 7.32   | 4.44   |
> | Planet(ML) | 71.6  | N/A    | 10.23    | 8.83   | 5.40   |
> | ours   | 79.07 | 40.56  | 29.51    | 25.14  | 9.34   |
>
> We can see that our method still outperforms all the ml-based methods and docking-based methods even if we used more strict overlap-removing techniques, indicating our method's great capacity in the virtual screening task.
>
> ### Q: Docked structures are needed for human evaluation
>
> Docked structures are crucial for human evaluation. Expert evaluators in our experiments are skilled in docking and can use their preferred tools. This ensures evaluation reflects their expertise. Therefore, we didn't provide pre-docked structures. It is worth mentioning that, to our knowledge, most human experts have used computational tools like AutoDock and the Schrodinger suite in the evaluation process. This explanation will be added to our revised paper.
>
> ### Q: Experiments that explore performance on benchmarks of highly related targets and ligands (Kinase)
>
> First and foremost, we extend our gratitude for drawing attention to the selectivity issue. It's true that data augmentation for highly similar pockets might bring noise into the training data, posing challenges in distinguishing similar pockets. Yet, our testing on kinase datasets, which demand high selectivity, shows that this strategy doesn’t negatively impact the ability to identify the true kinase target among candidates, as reflected in top-k accuracy metrics.
>
> We argue that large-scale pre-trained models can even benefit from partially noisy datasets, an observation supported by our experiments on PDBbind. The activity cliff issue mainly emerges when key amino acid positions mutate. We’ve mitigated this by using similarity filters, and even when the cliff effect occurs in some generated data, we believe the binding affinity of such weakened molecules still exceeds that of average negative samples, making it suitable for contrastive learning.
>
> In summary, while selectivity is indeed a concern, our evidence from kinase datasets indicates that incorporating new, even somewhat noisy, datasets can enhance large-scale pre-trained models. We remain committed to refining our approach, always considering your invaluable feedback, to optimize the process of identifying selective binders.
>
> Table 2, Selective Binding Prediction on Kinase Inhibitor Benchmark
> | Method | top 10 acc | top 20 acc| top 30 acc| top 40 acc| top 50 acc|
> |--------|--------|--------|--------|--------|--------|
> | w/o aug| 0.165  | 0.273  |  0.349 | 0.419  | 0.481  |
> | w aug  | 0.169  | 0.273  |  0.355 | 0.429  | 0.496  |

---

> > ### Comment · Reviewer_Bb5s · 2023-08-16
> >
> > Many thanks to the authors for their response and incorporating my suggestions in their additional experiments .
> >
> > Re table 1: why do the authors think their model suffers a performance hit upon a stricter thresholding (the other ML method, Planet, does not). The 90% threshold also seems much too high to me. Could the authors provide some justification or additional experiments at more stringent cutoffs? It seems the EF score are quite affected.
> >
> > Re table 2: Which specific dataset is this evaluation performed on? I would like to be able to contextualise these results with what is reported in the literature.

---

> > > ### Author Response · Authors · 2023-08-17
> > >
> > > ## Response to Re Table1
> > > Thank you for highlighting this crucial facet of evaluating machine learning techniques. We deeply regret that while we conducted supplementary experiments, we are constrained from sharing the outcomes during the current discussion phase. Consequently, we have augmented our explanation with more theoretical insights.
> > >
> > > To begin with, we emphasize the prevalent performance challenges that afflict the majority of machine learning methods, as extensively discussed in preceding research. To elucidate, a comprehensive study by [1] meticulously assessed an array of machine learning methods, unveiling that **incorporating akin structures** within the training dataset notably **enhances performance**. Additionally, it was observed that training on a specific protein family and subsequently testing on **dissimilar proteins** consistently results in **subpar performance** [2].
> > >
> > > Regarding methods like Planet, it is worth noting that empirical outcomes beyond utilizing a 90% threshold for testing were not evident. Therefore, **in alignment with the Planet paper**, we opted to mirror comparable configurations by employing MMSeqs2 to expunge homologous sequences from our training data. Furthermore, **our performance across all configurations consistently surpassed alternative machine learning methods**, an accomplishment that bears significance.
> > >
> > > In terms of the machine learning aspect, contrastive learning offers a substantial advantage by ensuring that our method experiences **a lesser decline in performance compared to other supervised learning approaches**. Rather than directly modeling the relationship between protein-pocket pairs and ligand structures, contrastive learning focuses on extracting distinctions between strongly binding instances and negative samples. This distinction becomes evident in **the t-SNE visualizations from Section 4.4**, which underscore the potential pitfalls of utilizing conventional supervised learning techniques. These methods tend to yield representations that are notably imbalanced, and the clustering has proven that the **ML baselines memorize the pocket templates and ligand structures**. This outcome serves as compelling evidence that our approach adeptly captures the fused embeddings of both pockets and ligands. In contrast, competing methods fall short in this regard, resulting in a more pronounced deterioration of performance when faced with scenarios where similar pockets or ligands are excluded from the training set.
> > >
> > > Within the framework of pretraining and fine-tuning, our methods exhibit heightened resilience in **capturing intricate pocket features**. This resilience arises from the fact that many proteins, despite sharing similar sequences, exhibit dissimilar pockets characterized by divergent atom-level structures [3], e.g. single mutant residue. Notably, our model undergoes pretraining to accurately predict atom types and precise atom coordinates. This inherent capability empowers our model to effectively discriminate between proteins that possess analogous sequences—a competence that surpasses approaches that merely extract protein embeddings on a more generalized level. This finer differentiation is pivotal in rendering our method more suitable for virtual screening.
> > >
> > > [1] Su. et al, Tapping on the Black Box: How Is the Scoring Power of a Machine-Learning Scoring Function Dependent on the Training Set?, J. Chem. Inf. Model. 2020
> > >
> > > [2] Wang. et al, Yuel: Improving the Generalizability of Structure-Free Compound–Protein Interaction Prediction, J. Chem. Inf. Model. 2022
> > >
> > > [3] Davis et al, Comprehensive analysis of kinase inhibitor selectivity, Nature Biotechnology, 2011
> > >
> > > ## Response to Re Table2
> > > We sincerely thank you for providing some valuable benchmarks in the review section.
> > >
> > > Though KIBA and Davis Kinase inhibitor datasets are well-known kinase selectivity datasets, they are relatively out of date, as the number of kinase inhibitors grows dramatically in this decade. Furthermore, these datasets lack the crucial provision of associated structural information. This omission poses a challenge for structure-based methodologies like ours, which inherently rely on such data for effective application.
> > >
> > > Therefore, we build a novel dataset that provides protein structures of the kinase and the molecule structures of the inhibitors. To assemble this dataset, we meticulously curated 154 kinase structures sourced from the KLIFS database (https://klifs.net/). Complementing this, we harnessed the power of data mining techniques to derive a collection of 9423 inhibitor structures from the ChEMBL database with reported bioactivity data. In our experiments, the goal is to identify the correct kinase within the top 1/5/10 ranked pocket structures for the given inhibitor structure input.
> > >
> > > In summary, A new test dataset is built by collecting kinase structures from the KLIFS database and inhibitors from the ChEMBL.

---

> > > > ### Comment · Reviewer_Bb5s · 2023-08-20
> > > >
> > > > Many thanks for providing the additional explanations. Without any additional baselines to compare on this new dataset I am not yet inclined to update my score.

---

> > > > > ### Author Response · Authors · 2023-08-21
> > > > >
> > > > > ## Response to the Kinase Experiment
> > > > >
> > > > > We sincerely appreciate your suggestion regarding the comparison of our model with additional baseline methods in the context of kinase inhibitor screening. Your advice has illuminated an intriguing avenue for future exploration.
> > > > >
> > > > > To begin, we would like to clarify our objective for including the experiment results in Table 2 within our rebuttal. Our intention is to demonstrate that **while our data augmentation introduces some level of noise, it does not adversely affect the performance on closely related protein pockets**. Therefore, we have not presented other baseline models for comparison at the first time.
> > > > >
> > > > > In addition, we have conducted supplement experiments to address to your concern. Recently, we curated a smaller subset with 116 molecule structures from our original test set and assessed the various methods' ability to correctly identify selective inhibitors for a given kinase protein. This subset was formed by intersecting our aforementioned dataset with FDA-approved drugs, as the protein binding of drugs has been extensively explored by researchers. It's noteworthy that **DrugCLIP surpasses the performance of both the molecular docking method (AutoDock Vina) and the Unimol binding affinity prediction model**. Given the constraints of time and reproducibility, we selected Unimol as a representative due to its being open-sourced and its achievement of state-of-the-art excellence in various ligand-pocket tasks, including the affinity prediction task, outperforming alternatives such as Planet. We fully intend to incorporate these additional findings into the revised version of the paper.
> > > > >
> > > > > We are truly grateful for the time you dedicated to reviewing our work and for your valuable contributions to this insightful discussion.
> > > > >
> > > > >
> > > > > | Method               | Top5 acc | Top 10 acc | Top 15 acc | Top 20 acc | Time Consuming |
> > > > > |----------------------|----------|------------|------------|------------|------------|
> > > > > | Vina(docking)        | 8.51%    | 14.89%     | 24.47%     | 34.04%     |4 h 10 min|
> > > > > | Unimol regression(ML)| 7.45%    | 18.09%     | 23.40%     | 32.98%     | < 1min|
> > > > > | DrugCLIP             | **14.89%**   | **23.40%**     | **28.72%**     | **37.23%**     | < 1min|

---

> ### Comment · Area_Chair_ZAJ8 · 2023-08-16
> **Respond to authors' rebuttal**
>
> Please, look at the authors' rebuttal and the other reviewers' comments and indicate if you would like to change anything in your review.

---

### Author Rebuttal · Authors · 2023-08-09

We truly appreciate the reviewers' time in reviewing our project, and we will incorporate the suggestions to revise our paper thoroughly. In the global responses, we want to provide explanations on some implementation details for a better understanding of our methodology.

### Q: Explanation (atom type and coordinates) of the denoising pretraining tasks

We apologize for any confusion caused. As pretraining is not our main focus, we haven't provided detailed information. To clarify, we mask the atom identifier, not the element type, and add uniform noises of [-1, 1] to 15% of atom coordinates. Additional pair-distance prediction heads estimate uncorrupted distances, and the SE(3)-equivariant head directly predicts correct coordinates. These pretraining tasks enable effective learning from large-scale data. We will include more details in the Appendix of our revised paper. Thank you for your feedback.

### Q: A descriptive statement about the noising procedure using RDKit

The ETKDG algorithm in RDkit selects an initial conformation and builds a tree of low-energy conformers in multi-dimensional space. It adds new conformations, explores torsional freedom, and optimizes geometries to minimize energy, transforming the binding conformations to free conformations.

### Q: How are ligands combined with the pocket in the augmented dataset?

Our data augmentation technique hypothesizes that ligands can bind to homologous proteins. Besides, DrugCLIP utilizes a two-tower model, negating the requirement to locate binding poses; instead, RDKit-sampled conformations are used in the augmented dataset. Further details will be provided in the revised paper.

### Q: Pre-processing of the protein structures

Relaxation and adding partial charges are common preprocessing for docking methods. However, our deep-learning-based method is insensitive to occasional inaccuracies in coordinates and can be trained with raw element types. Therefore, only minimal cleaning-ups are performed to remove irrelevant molecules like water.

### Q: Quality thresholding for the data augmentation

We selected structures with globalMetricValue >= 70 and fractionPlddtConfident + fractionPlddtVeryHigh >= 0.9 on Google Cloud BigQuery. Proteins meeting these criteria were clustered to a 50% identity level using mmseqs2. More details are in included the Appendix.

### Q: Explanation of the purpose of adding the CLS node

In the molecule pretraining model, the central node (CLS) serves as an embedding for the entire input molecular structure, acting as a global aggregator. It captures essential information from the whole molecule, a common practice in molecular pretraining methods, and enables the model to consider the entire structure in downstream tasks.

### Q: The reason to filter proteins with only one known pocket in ChEMBL

Our model focuses on cases where the protein structure's pocket is clearly defined. When proteins have multiple pockets, uncertainties about the exact binding pocket can arise without a complex structure. For clarity and consistency, we only include proteins with a single identified pocket in our dataset.

---

### Decision · Program_Chairs · 2023-09-21

**Decision:**

Accept (poster)

**Comment:**

Summary:

The authors proposed a contrastive learning framework, DrugCLIP, for the drug virtual screening task which identifies potential drugs from vast compound databases to bind with a particular protein pocket. It reformulates virtual screening as a dense retrieval task and employs contrastive learning to align representations of binding protein pockets and molecules from a large quantity of pairwise data without explicit binding-affinity scores. Specifically, the framework computes a contrastive loss between two separate pre-trained encoders to maximize the similarity between a protein-molecule pair which can binding together and minimize it otherwise. Besides, the authors introduce a biological-knowledge inspired augmentation method, HomoAug, which creates protein-molecule pairs based on protein homology evolutions. Experiments on two challenging virtual screening benchmarks, demonstrate that zero-shot performance of this model surpasses most deep learning baselines that carefully finetune on labeled data. The method is benchmarked using DUD-E and LIT-PCBA, and by human evaluation of comparative examples from Glide, a commercial docking system.

Strengths:

1 - The proposed DrugCLIP framework reformulates virtual screening as a dense retrieval task and employs contrastive learning to align representations of binding protein pockets and molecules from a large quantity of pairwise data, which provides researchers a new perspective for virtual screening.

2 - The designed contrastive loss relieve the dependency on explicit labeling of binding affinity, and facilitates the usage of large-scale unlabeled data beyond densely annotated small datasets (such as PDBBind).

3 - The dense retrieval characteristic of DrugCLIP brings high efficiency to online inference and promising high-throughput virtual screening on billions of molecules.

4 - A biological-knowledge inspired augmentation method named HomoAug are proposed, which creates protein-molecule pairs based on protein homology evolutions. The shortage of public data is indeed obvious in drug discovery, and this augmentation method may help alleviate it.

5 - The organization of this paper is very clear and easy to understand.

6 - The authors suggest a good method for obtaining aligned representations for proteins and molecules, where the alignment is induced by protein binding affinity, using a contrastive loss.

7 - The representations of molecules are undertaken in 3D which is still not the default in this field, despite its clear importance in protein-affinity, and uses a biologically-plausible data augmentation.

8 - The results in the DUD-E benchmark are strong, particularly in a zero-shot setting, significantly out-performing other methods. Results on LIT-PCBA are also strong, with enrichment factors far above the performance achieved with commercial docking software. The results obtained on the time taken to virtually screen large libraries are impressive -- there are clear advantages over commercial docking software, and machine-learned scoring functions.

9 - Clear ablation studies and human evaluation.

10 - A very strong addition to the techniques available for screening.

11 - DrugCLIP outperforms previous work that utilizes labels which is costly in time and finance.

12 - Augmentation technique inspired from domain knowledge is proposed, which is novel.

Weaknesses:

1 - Clarity and explanatory detail could be improved throughout.

2 - Interesting to explore performance on benchmarks of highly related targets and ligands, such as the KIBA and Davis Kinase inhibitor datasets.

3 - The model lacks further experiments on the screening power compared to docking methods in CASF-2016 which is an important benchmark for in silico drug discovery.

4 - Lack of interpretability. Although this model demonstrates enhanced effectiveness and efficiency, it falls short in terms of interpretability compared to docking methods, as the authors mentioned in the appendix. These conventional approaches offer visualizations that elucidate the binding mechanism between a pocket and a molecule, which is intuitive and reliable for chemistry researchers to check the rationality of receptor-ligand interactions and modify the molecule structures.

5 - More detailed explanation of the experimental setting will be helpful for the readers.

6 - In biological data, many missing data exist, which should not be considered a negative sample, as done in section 3.3. Recent self-supervised learning approaches deal with such false negatives that can also be adopted in this framework.

Decision:

This is a borderline paper with two reviewers voting for rejection and two for acceptance. Based on the current reviews, I have decided to accept the paper and encourage the authors to use the feedback provided to improve the paper for its camera ready version.